# Chitosan as a Control Tool for Insect Pest Management: A Review

**DOI:** 10.3390/insects14120949

**Published:** 2023-12-15

**Authors:** Linda Abenaim, Barbara Conti

**Affiliations:** Department of Agriculture, Food and Environment, University of Pisa, Via del Borghetto 80, 56124 Pisa, Italy; barbara.conti@unipi.it

**Keywords:** biopolymer, chitin, insect control, chitosan formulations, chitosan nanoparticles

## Abstract

**Simple Summary:**

Chitosan is a biopolymer derived from chitin that has gained much attention due to its biological activities. Chitosan can be produced by the exoskeleton of arthropods (crustaceans and insects) and the structural membranes and spores of fungi. Its application has spread to many sectors, including the pharmaceutical, medical, veterinary, food, and agricultural sectors. In the latter one, chitosan is applied to improve the interaction between plants and microorganisms and the metabolisms of plants, fruits, and germination. In addition, chitosan has been demonstrated to enhance the availability and stabilization of insecticides and essential oils. Several chitosan formulations have been studied as tools for insect pest control. This review aims to focus on the role of chitosan as a pest management tool by examining all possible chitosan formulations.

**Abstract:**

Chitosan, a polysaccharide derived from the deacetylation of chitin, is a versatile and eco-friendly biopolymer with several applications. Chitosan is recognized for its biodegradability, biocompatibility, and non-toxicity, beyond its antimicrobial, antioxidant, and antitumoral activities. Thanks to its properties, chitosan is used in many fields including medicine, pharmacy, cosmetics, textile, nutrition, and agriculture. This review focuses on chitosan’s role as a tool in insect pest control, particularly for agriculture, foodstuff, and public health pests. Different formulations, including plain chitosan, chitosan coating, chitosan with nematodes, chitosan’s modifications, and chitosan nanoparticles, are explored. Biological assays using these formulations highlighted the use of chitosan–essential oil nanoparticles as an effective tool for pest control, due to their enhanced mobility and essential oils’ prolonged release over time. Chitosan’s derivatives with alkyl, benzyl, and acyl groups showed good activity against insect pests due to improved solubility and enhanced activity compared to plain chitosan. Thus, the purpose of this review is to provide the reader with updated information concerning the use and potential applications of chitosan formulations as pest control tools.

## 1. Introduction

In recent decades, research has increasingly sought alternative, biodegradable, and ecological materials for various scientific applications. Biopolymers such as cellulose, starch, gelatines, and chitosan have gained much attention due to their valuable characteristics, including biodegradability, biocompatibility, and non-toxicity for humans [1,2]. Chitosan is the most important derivative of chitin. It was discovered in mushrooms by Henri Braconnot back in 1811, who named it “fungine”. Later, in 1821, Auguste Odier isolated the same compound from beetle cuticles and named it “chitin” [3,4].

Chitin was, thus, the first polysaccharide identified by man, preceding cellulose by about 30 years, and the most abundant polymer after the latter. Indeed, chitin is the most abundant component in the exoskeleton of arthropods and in the structural membranes and spores of fungi [2,5]. Shrimps and crabs, characterized by a high chitin content (around 20–30%), are the most common commercial sources of this polysaccharide, although their use as a source of chitin has been much debated recently [6,7].

As the global demand for chitosan is increasing, with an annual growth rate of 15.4%, there is a need for alternative sources to meet the market demand. In recent years, research has also focused on the study of insects, characterized by 10–15% of chitin, as an alternative source of the latter [8]. The extraction of chitin from insects is also advantageous in terms of the rearing method, substance consumption, and time of production [9]. The most investigated insect species for chitosan production are *Hermetia illucens* (Linnaeus, 1758) (Diptera Stratiomyidae), *Tenebrio molitor* (Linnaeus, 1758) (Coleoptera Tenebrionidae), and *Alphitobius diaperinus* (Panzer, 1797) (Coleoptera Tenebrionidae). Chitosan production from chitin, regardless of the source, is mainly characterized by three steps of extraction, as follows: demineralization, deproteinization, and deacetylation [2,9]. However, different conditions during the extraction process, such as concentration of reagents, time and temperature, pH, presence, or absence of metal cations, pKa, molecular weight, and degree of deacetylation, may affect the properties of the biopolymer [10].

Chitosan has well-known antimicrobial, antioxidant, and antitumoral properties [10,11,12], and its application has spread to many sectors, including the pharmaceutical, medical, veterinary, food, and agricultural sectors. In the pharmaceutical industry, chitosan is used in drug formulas to prevent stomach irritation thanks to its antacid and antiulcer properties [2]. In the cosmetic industry, it is used for hair and skin treatments; in medicine, it is used for orthopaedic and periodontal implants, and, in the textile field, it is employed for its anti-static properties and in water purification [1,2,13]. In animal nutrition, chitosan is used as an antimicrobial to protect feed from *Escherichia coli* and *Salmonella* spp. contamination, while in the food industry it is used for food packaging [14]. In agriculture, chitosan is applied to soil, promoting symbiotic interaction between plants and microorganisms and improving the metabolisms of fruits and plants and germination [2,13]. 

## 2. Use of Chitosan against Insects

Chitosan can enhance the availability and stabilization of some insecticides or botanicals, while some chitosan derivatives have demonstrated insecticide activity towards various agricultural pests [12,15]. For this reason, chitosan is currently authorised in the EU as an active and non-toxic basic substance for plant protection as an environmentally friendly and safe alternative to synthetic pesticides (Regulation EC No 1107/2009) [16,17]. Various formulations of chitosan, supplemented with natural or synthetic insecticides, were studied as insect pest control tools in agriculture and public health but also as instruments for protecting foodstuffs [18,19,20,21]. In particular, the polymer has been used in a variety of formulations, including a plain chitosan formulation sprayed on leaves, wood, and paper, chitosan derivatives, chitosan–metal complexes, chitosan nanoparticles loaded with essential oils (EOs) and insecticides, chitosan coating, and chitosan formulations with nematodes.

The use of chitosan in different formulations has been the subject of many studies. Thus, the purpose of this review is to provide the reader with updated information concerning the use and potential applications of chitosan formulations, summarized in Table 1, as pest control tools.

### 2.1. Plain Chitosan

Recently, some papers have pointed to chitosan as a tool for controlling insect pests in agriculture, foodstuffs, and public health. However, the insecticidal activity seems related to the antimicrobial effect of the chitosan that impairs the function of the insects’ gut microbiota.

In fact, Stoffolano and co-authors reported that a 2% chitosan solution incorporated in an artificial diet induced a statistically significant reduction in the survival period of *Musca domestica* (Linnaeus, 1758) (Diptera Muscidae) from 13 to 4 days in the control diet. This trend is also confirmed for *Tabanus nigrovittatus* (Macquart, 1847) (Diptera Tabanidae) (from 16 to 4.5 days) and for *Phormia regina* (Meigen, 1826) (Diptera Calliphoridae) (from 24 to 6 days). The insect mortality shown in this study appears to be due to physiological and structural changes in the midgut tract, even if the mode of action has not yet been clarified [20].

Even Muryeti and co-authors [59] reported that milled paper supplemented with chitosan and administered to termites showed an increase in their mortality compared to the control (80 and 45% mortality, respectively). This seems to be due to the fact that chitosan paper interferes with termite feeding, disturbing the ability of symbiotic protists in the termites’ digestive tract. This prompted the research to discover the effect of chitosan on the diversity and number of protists in termites’ guts. Chitosan was also reported as being potentially effective in preserving wood against termites [60]. In particular, in the work of Raji et al. [24], chitosan-treated wood showed a statistically significant mortality against subterranean termites of more than 94% for *Reticulitermes flavipes* (Kollar, 1837) (Blattodea, Rhinotermitidae) when exposed for 28 days to the chitosan (2% of concentration). On the contrary, for the other species tested, such as *R. virginicus,* more than 90% mortality was obtained at the lowest concentrations of chitosan (0.5%). Moreover, in Telmadarrehei et al. [23], the effect of chitosan wood treatment was investigated against the protists of *Reticulitermes virginicus* (Banks, 1907) (Blattodea, Rhinotermitidae). From the results, termites of the control group showed ten protist species in the hindgut, while in the specimens treated with chitosan the presence of protists was reduced to only two species.

### 2.2. Chitosan Coating

Because of its film-forming and biochemical capabilities, chitosan is a good preservative for fresh fruits and vegetables, according to numerous studies [61,62]. Furthermore, besides its antimicrobial activity, unlike other coating materials, plain chitosan displays antifungal activity against a variety of fungi, including *Aspergillus niger, Botrytis cinetica,* and *Rhizopus stolonifer* [63,64].

Plain chitosan coating is also safe for humans [65] and semi-permeable and has the potential to change the internal atmosphere of food, e.g., by minimizing transpiration losses, delaying spoilage, and, thus, extending the shelf-life of various foods [66,67].

In agriculture, plain chitosan coatings have also been tested for seed and fruit treatment for plant growth stimulation and disease control [61,68]. In particular, some studies have reported that chitosan coating repels insects and inhibits the settling of certain developmental instars. Indeed, in Salvador-Figueroa et al. [25] and Limon et al. [26], a chitosan coating on mango fruit inhibited the development of eggs and larvae of Mexican fruit flies, *Anastrepha ludens* (Loew, 1873) and *Anastrepha obliqua* (Macquart, 1835) (Diptera Tephritidae). In both cases, the chitosan coating significantly slowed the ripening rate and reduced the weight of the fruit. Regarding the mode of action, the authors [25,26] indicated an increase in the concentration of phenolic compounds and in the gas exchange of the fruit, both responsible for the inhibition of the development of eggs and larvae. In addition, due to the different colours and penetrability of the treated fruits, the chitosan coating behaved as a barrier, reducing the fruits’ attractiveness to pests and inhibiting the insects’ oviposition [69].

### 2.3. Chitosan Coating with Essential Oils

In the relevant literature from the last 30 years, essential oils (EOs) have been proposed as eco-friendly insecticides and/or repellents for insects [70,71,72,73,74,75] even in different formulations [76]. However, they present some use limitations due to their pronounced volatility. Chitosan nanoparticles loaded with EOs can improve their bioactivity and stability by decreasing their volatility. Indeed, EOs, known for their insecticidal and insect-repellent activity, added to chitosan’s results in increases in their bioactivity. In fact, by applying on the fruit of *Psidium guajava* (common guava) the film produced by adding 2% of citronella EOs and, as scaffolding excipients, polyethylene glycol (PEG) and carboxymethylcellulose (CMC), the oviposition of *Bactrocera carambolae* (Drew & Hancock, 1994) (Diptera Tephritidae) was significantly inhibited by 85% when compared to the control [28]. In a similar work [27], cabbage seeds coated with chitosan, enriched with jasmonic acid, induced a 57% preimaginal mortality of *Plutella xylostella* (Linnaeus, 1758) (Lepidoptera Plutellidae) and *Myzus persicae* (Sulzer, 1776) (Hemiptera, Aphididae) before reaching adulthood, preventing adult emergence.

The coating of chitosan loaded with different EOs has demonstrated antifeedant and anti-toxic properties, enhancing EOs’ effectiveness, durability, and availability. In fact, in Ascrizzi et al. [30], the chitosan coating film, obtained with the addition of *Ferulago campestis* EO, tested on bean seeds showed a statistically significant dose-dependent repellent effect (93.3% repellence at the highest concentration 57.7 µL/L of air) against the seed pest *Acanthoscelides obtectus* (Say, 131) (Coleoptera Bruchidae), without leading to adverse effects on bean plant germination and growth but resulting, at the same time, effective in suppressing weed seed germination in vitro.

For the protection of fresh foods, chitosan loaded with EOs has been tested as a protective coating to preserve meat against the oviposition of *Calliphora vomitoria* (Linnaeus, 1758) (Diptera Calliphoridae) [29]. From the results, chitosan at a 1% loaded with 0.1% of *L. nobilis* and *P. nigrum* EOs (chosen following the sensory profile analysis) showed a protective effect on meat with statistically significant oviposition reduction in *C. vomitoria* of 84.9 and 93.3%, respectively, for the two EOs compared to the control. Moreover, the treatment was effective in preserving the quality and shelf-life of the meat, delaying meat desiccation and lipid peroxidation by 7 days.

Hossain et al. [31] developed a chitosan coating containing eight different EOs also in combination with ionizing radiation (specifically gamma radiation at 100 and 300 Gy). The eucalyptus and tea tree EOs loaded in a chitosan coating in rice showed the most effectiveness towards *Sitophilus oryzae* (Linnaeus, 1758) (Coleoptera Curculionidae), causing 100% mortality at the lowest concentrations of 0.2 µL/mL for an incubation period between 24 and 48 h. With the combination of gamma radiation, the mortality of insects was 100% even after 14 days of treatment, providing complete pest control for the food’s packaging.

### 2.4. Chitosan Coating in Active Packaging

Chitosan can also be used as a secondary compound in active packaging systems to control foodstuff insect pests [77]. Active packaging refers to the incorporation of some active compounds into the packaging material, in this case chitosan-based, to maintain the quality and extend the shelf-life of food products. Recently, De Fàtima Silva et al. [32] developed a sustainable active packaging material based on the coating of a cardboard surface with chitosan and lemongrass EO, and they evaluated the insecticidal activity towards *Sitophilus zeamais* (Motschulsky, 1885) (Coleoptera, Curculionidae). In particular, in this formulation, chitosan, thanks to its amino groups, formed hydrogen bonds with the hydroxyl groups of the cardboard cellulose, providing good strength for the paper and improving the barrier properties of the paper matrix whilst conserving biodegradability, while the EO contributed to an anti-insect effect. In fact, the active packaging material was fully effective against the maize weevil attack, with 100% toxicity observed even after 360 h from the treatment. Moreover, the air permeability and water absorption capacity of the active packaging material were improved compared to the uncoated packaging.

### 2.5. Chitosan and Nematodes

Entomopathogenic nematodes (EPNs) are used in traditional, conservation, and supplemental biological control programs [78,79]. Most of the existing applied research concerns their potential as biological control agents [80,81,82]. According to Abbas et al. [83], *Steinernema carpocapsae* is the most available, versatile, and effective EPN, mutualistically associated with *Xenorhabdus nematophila*, against *Rhynchophorus ferrugineus* (Oliver, 1790) (Coleoptera Curculionidae). Recent laboratory and semi-field investigations have demonstrated the efficacy of combining *S. carpocapsae* with chitosan against *R. ferrugineus* on a *Phoenix canariensis* palm [33]. According to these results, in both Llácer et al. [33] and Dembilio et al. [34], chitosan activated the defence mechanisms in the palm, increasing lignification and root development. The chitosan formulation with nematodes, already patented in Spain [84], showed interesting results through curative assays (81.3% larval mortality after 28 days of application) and preventive assays (98.2% immature stage infestation after 15 days of application) [33]. Consistent with these results, Dembilio et al. [34] found 99.7% mortality of the immature stages of *R. ferrugineus* in a palm stipe treated with *S. carpocapsae* formulated with chitosan.

### 2.6. Chitosan Chemical Modification

Since chitosan itself has no toxic properties, its use against harmful insects is usually as an additive to other toxic substances. Nevertheless, its use in water is often limited due to its intermolecular and intramolecular hydrogen bonds, which are highly crystalline, hindering the material’s ability to interact with water molecules [85]. For this reason, research has turned to the study of new derivatives and chemical modifications of chitosan that can compensate for this lack of solubility and toxicity and improve its physicochemical properties in order to broaden its applications. An example is the chitosan oligosaccharide (COS) (Figure 1), which is produced using various methods, including enzymatic and acid hydrolysis, which convert chitosan to low molecular weight units and provide it with the ability to be soluble in aqueous solutions. These properties of chitosan oligosaccharides also improve the biological activities of chitosan, as reported in numerous studies. In particular, the properties of chitosan derivatives have been studied against certain insects harmful to agriculture [86,87,88]. Zhang et al. [35] reported the insecticidal activity of a chitosan oligosaccharide solution sprayed on the leaves of plants attacked by some of the most common pests in agriculture, including the Lepidoptera *Helicoverpa armigera* (Hübner, 1808) (Noctuidae) and *P. xylostella* and the Hemiptera Aphididae *Rhopalosiphum padi* (Linnaeus, 1758)), *Sitobion avenae* (Fabricius, 1775), *Metopolophium dirhodum* (Walker, 1849), *M. persicae*, *Hyalopterus pruni* (Geoffroy, 1762), and *Aphis gossypii* (Glover,1877). The results obtained using the chitosan solution showed an insecticidal activity of 40% for *H. armigera* and 72% for *P. xylostella* after 72 h. The treatment was proved to also be effective against aphids, especially against *H. pruni*, (93% mortality).

Other authors have demonstrated that chitosan derivatives, with the introduction of functional groups at the N- and O- positions (Figure 2), show improved biodegradability and chemical solubility in water or organic solvents as well as enhance chitosan’s antimicrobial properties [89].

The new chitosan derivatives N-alkyl chitosan (NAC), N-benzyl chitosan (NBC), and O-(acyl) chitosan (OAC) were administrated in an artificial diet to the larvae of the Lepidoptera Noctuidae *Spodoptera littoralis* (Boisduval, 1833). Among the chitosan derivatives tested, N-(2-chloro-6-fluorbenzyl) chitosan was the most effective against *S. littoralis* larvae (LC_50_ of 0.32 g/kg and LC_100_ of 0.625 g/kg of diet), even if not statistically significant [36]. Moreover, N-(propyl) chitosan, N-(undecanyl) chitosan, N-(3-phenyl propyl) chitosan [37], and O-(decanoyl) chitosan [38] showed larval growth inhibition with larval weight reductions of 76, 66, 65, and 64%, respectively, after 4 days of administration, even if the results were not statistically significant.

Moreover, chitosan complexes with metals modify both the activity and the physiochemical characteristics of plain chitosan [90]. As a chelating agent, chitosan may easily form complexes with transition or heavy metals (Figure 3). Metal ions such as Ag^+^, Cu^2+^, Zn^2+^, and others are known to be antimicrobial agents [91]. In this regard, some studies have focused on the antibacterial properties of chitosan–metal complexes. In fact, in Wang et al. [92], chitosan–metal complexes exhibited pronounced antimicrobial activities indicating that the inhibitory effect of the chitosan–metal complexes depended on the property of the metal ions and their molecular weight. Knowledge of the mode of action of these complexes against insects has also been considered of great interest. Metal complexes like AgNO_3_, CuSO_4_-5H_2_O, NiCl_2_-6H_2_O, and HgCl_2_ were added to a solution of different molecular weights of chitosan to obtain some chitosan–metal complexes that were then tested against *S. littoralis* and *Aphis nerii* (Boyer de Fonscolombe, 1841) (Hemiptera Aphididae) [39]. After 7 days of administration to *S. littoralis* larvae, the chitosan at a low molecular weight (2.27 × 10^5^ g/mol) incorporated in the artificial diet exhibited 50% larval mortality, while the Chitosan–Ni and Chitosan–Hg complexes exhibited significantly higher larval mortality (93 and 83%, respectively) when compared to the control but also to the other chitosan–metal complexes. Instead, for *A. nerii*, the three solutions of chitosan at different molecular weights (2.27 × 10^5^, 3.60 × 10^5^, and 5.97 × 10^5^ g/mol) all had significant high efficacy (96, 87, and 100%, respectively) when compared to the chitosan–metal complexes. However, against *A. nerii*, the chitosan–Cu complex was the statistically significant most effective chitosan–metal complex, resulting in a 94% mortality at 48 h.

### 2.7. Chitosan Nanoparticles

Polymeric nanoparticles have recently gained popularity in a wide variety of contexts, mainly in the agricultural sector [15,93,94]. In particular, chitosan nanoparticles are being investigated as carriers of active substances, thanks to their controlled delivery systems, and for the stabilization of biological components such as proteins, peptides, or genetic material [95].

Ionotropic gelation, polyelectrolyte complex, microemulsion, emulsion solvent diffusion, and inverse micellar techniques are the systems employed to synthesize chitosan nanoparticles. However, the most common techniques are the first two mentioned above [96].

According to their morphology, nanoparticles are divided into two major groups: nanospheres, solid structures with a homogeneous matrix in which the active ingredient uniformly coats them externally, and nanocapsules, which are hollow structures with a polymer membrane and an inner core containing the active ingredient [97]

With diameters ranging from 1 to 100 nm, the size, shape, surface area, and area/volume ratio of nanoparticles are crucial characteristics for their successful use in pest control. The smaller size of nanoparticles (Figure 4) ensures better coverage, permeability, and bioavailability of the active ingredient, and it also enhances water solubility for otherwise-insoluble active ingredients, formulation stability, slow-release capability, resistance to degradation, mobility, and increased insecticidal activity. Consequently, nanometre-sized particles are the most commonly utilized in chitosan formulations [93].

One of the most common uses of this formulation is loading chitosan nanoparticles with EOs to increase the bioactivity and stability of the oils. In Rajkumar et al. [20], chitosan nanoparticles loaded with *Piper nigrum* EO were evaluated against *Tribolium castaneum* (Herbst, 1797) (Coleoptera Tenebrionidae) and *S. oryzae*. In fumigation toxicity tests, carried out using an impregnated paper assay [40], chitosan nanoparticles loaded with *P. nigrum* EO showed statistically significant prominent mortality after 24 h with LC_50_ of 25.03 and 29.02 μL/L air for *S. oryzae* and *T. castaneum*, respectively, compared to the pure EO (LC_50_ of 48.97 and 55.77 μL/L air, respectively). Moreover, in antifeedant bioassays, the nanoparticles showed greater wheat grain protection from *S. oryzae* and *T. castaneum* attack (0% of damages observed for both species) when compared to the control (74% damages observed for *S. oryzae* and 86% for *T. castaneum*). The toxicity of chitosan nanoparticles loaded with EOs was confirmed in a later work by the same authors [21], in which chitosan nanoparticles loaded with peppermint EO (administered using the same methodology) showed a similar toxic effect with LC_50_ of 28.61 and 34.79 μL/L air against *S. oryzae* and *T. castaneum*, respectively, also resulting more effective when compared to the control (LC_50_ of 56.48 and 62.94 μL/L air, respectively). Similarly, chitosan nanoparticles loaded with the EO of *Melissa officinalis* showed higher statistically significant toxicity against *T. castaneum* by fumigation (LC_50_ of 0.048 μL/mL air) than the control (LC_50_ of 0.071 μL/mL air). The antifeeding potential of chitosan nanoparticles loaded with *M. officinalis* EO was tested in dedicated bioassays, which revealed that *T. castaneum* was deterred from feeding on the treated flour discs (maximum deterrence of 80%) [41].

Moreover, chitosan nanoparticles contribute to a slow release of EOs and prolong their effect over time, compared to the neat product. In fact, in Soltani and co-authors [42], fumigation tests with nanocapsules of chitosan loaded with *Rosmarinus officinalis* EO showed an extension of mortality over time of *T. castaneum* (20% mortality after 60 days compared to the pure EO after the same time, at 2.1%). These results were confirmed by Soltani and co-authors [43,44], who reported that the rosemary oil nanoencapsulated by chitosan exhibited 82% and 50.7% mortality after 60 days against *Oryzaephilus surinamensis* (Linnaeus, 1758) (Coleoptera, Silvanidae) and *Carpophilus hemipterus* (Linnaeus, 1758) (Coleoptera, Nitidulidae) compared to the neat EO (62% and 19% mortality, respectively). Following these results, the EO nanoencapsulation in chitosan was more effective than the neat EO even over time.

Chitosan nanoparticles loaded with EOs were also tested against mosquitoes, with good results. In fact, *Elettaria cardamomum* and *Cinnamomum zeylanicum* EOs loaded in chitosan nanoparticles showed significant larvicidal activity, after 24 h, against *Anopheles stephensi* (Liston, 1901) (Diptera Culicidae), with LC_50_ values of 7.58 and 2.98 μg/mL, respectively [45]. In other studies, chitosan nanoparticles loaded with the EOs of *Geranium maculatum* and *Citrus bergamia* showed statistically significant larvicidal activity against *Culex pipiens* (Linnaeus, 1758) (Diptera Culicidae) with LC_50_ of 22.63 and 38.52 ppm, respectively, and also mortality over time. In detail, mortality was more than 50% even 5 days after treatment [46].

Similar results were obtained with chitosan nanoparticles loaded with *Lippia sidoides* EO, which showed significant high larvicidal activity after 3 days (specifically 85% after 24 h and 92% after 48 and 72 h) against *Ae. aegypti* [47]. Likewise, in another study, chitosan nanoparticles loaded with *Siparuna guianensis* EO, at various concentrations, induced significant larval mortality of *Ae. aegypti* in a time- and dose-dependent way [19]. In this case, first, instar mosquito larvae were changed daily for 2 weeks to observe the effect of the nanoparticles over time. The nanoparticles prepared with a 1:2 ratio of chitosan/EO showed 100% mortality after 2 days at the lowest concentration (0.83 mg/mL), while the nanoparticles with the ratio 1:1 of chitosan/EO at the highest concentration (6.67 mg/mL) showed more than 80% mortality even 2 weeks after treatment.

In some cases, chitosan nanoparticles loaded with EOs or their compounds can be attractive to *Bemisia tabaci* (Gennadius, 1889) (Hemiptera, Aleyrodidae), with a prolonged effect on pests, as reported by De Olivera et al. [49]. In the latter’s work, chitosan nanoparticles loaded with geraniol (5 mg/mL), one of the main components of some EOs, produced a stable attractive effect even after 60 days. On the contrary, Ferreira et al. [19], obtained a high toxicity at 6.67 mg/mL of the OE of *S. guianensis* loaded in chitosan nanoparticles. In fact, it has been shown that the bioactivity of EOs is vectorial, i.e., it changes sign depending on the EO and on the dose used. The nanoencapsulation of limited amounts of EOs, which also results in a controlled release of their volatile components, can, therefore, turn repellence into attractiveness [98,99,100]. These results suggest that nanoencapsulation of a small amount of EOs in chitosan could be used in potential traps.

Vegetable plant extracts possess natural protective properties against pests [101]. Moreover, when combined with chitosan nanoparticles, the insecticidal effect of plant extracts appears to be enhanced. In fact, the addition of chitosan nanoparticles loaded with *Nerium oleander* leaf extract to an artificial diet resulted in a statistically significant larvicidal effect against *M. domestica* 48 h after exposure. The LC_50_ was 0.64 ppm, and there was a reduction in the pupation rate and adult emergence of 27% and 60%, respectively [48].

Chitosan nanoparticles have been also applied as nanocarriers for agrochemicals [102]. In this regard, the nanoencapsulation of Poneem (a botanical pesticide containing neem oil, Karanja oil, azadirachtin, and Karanjin) in chitosan was tested against *H. amigera* larvae [18]. The chitosan nanoparticles were obtained by tripolyphosphate (TPP) or glutaraldehyde (GLA), commonly used as crosslinking agents in chitosan nanoparticle preparation, increasing their stability. The chitosan–TPP–Poneem and chitosan–GLA–Poneem obtained showed significant antifeedant activity of larvae up to 88.5 and 72.3%, respectively, and larvicidal activity up to 90.2% for chitosan–TPP–Poneem and 87.5% for chitosan–GLA–Poneem. Like some of the botanicals contained in the agrochemicals, these nanoparticles induced growth and developmental abnormalities in the larvae and pupae.

Moreover, the chitosan nanoparticles loaded with insecticidal metabolites extracted from the fungal biocontrol agent *Nomuraea rileyi* and sprayed on leaves showed up to 99% larvicidal activity for the fourth larval stage of *Spodoptera litura* (Fabricius, 1775) (Lepidoptera Noctuidae) after 48 h from the treatment [40].

Chitosan nanoparticles have been also loaded with commercial insecticides (Spinosad and Permethrin), showing a significant dose-dependent ovicide effect against *Drosophila melanogaster* (Meigen, 1830) (Diptera Drosophilidae) [51].

Even chitosan-g-poly acrylic acid nanoparticles, obtained by grafting an acrylic acid monomer onto chitosan and administrated on castor leaves, negatively influence the percentage of *A. gossypii* growth performance and adult emergence (77.8% and 75% decrease in growth and adult emergence compared to the control) [52].

Instead, the nanochitosan-g-poly acrylic acid tested against *Cassida vittata* (Villers, 1789) (Coleoptera Chrysomelidae) showed a statistically significant decrease in oviposition compared to plain chitosan (3 ± 8.9 and 266 ± 8.7 eggs, respectively). The new product also induced 100% egg mortality and 91% larval mortality [53].

Self-assembled chitosan nanoparticles with fatty myristic acid (MA–chitosan nanoparticles) loaded with *Cuminum cyminum* EO were tested against foodstuff insect pests. The self-assembled compound showed, after 48 h, significant mortality of 100% of the *Sitophilus granarius* (Linnaeus, 1758) (Coleoptera Curculionidae) and *Tribolium confusum* (Jaquelin du Val, 1863) (Coleoptera Tenebrionidae) adults, with efficacy persistence over time (50% mortality after 12 days from exposure for *S. granarius* and 50% after 24 days for *T. confusum*) [54]. These results have been confirmed in a similar study by the same authors, where MA–chitosan nanogel loaded with *Carum copticum* EO was tested against the same pest species [55]. Also, in this last case, the mortality was maintained over time, with 89% and 80% mortality after 48 h for *S. granaries* and *T. confusum*, respectively, and 20% mortality after 12 days for *S. granaries* and 40% after 24 days for *T. confusum*.

Recently, chitosan nanoparticles have gained popularity as a component of small interfering RNA (siRNA) and double-strand RNA (dsRNA) formulations [15]. The cationic nature of chitosan enables the application of chitosan nanoparticles for RNAi, a non-invasive mechanism by which specific RNA fragments (dsRNA and siRNA) are able to silence gene expression in mosquito larvae by feeding [56,57,58]. In particular, RNAi technology using chitosan nanoparticles has improved its efficacy as a pest control tool, particularly for *Anopheles gambie* (Giles, 1902) (Diptera Culicidae) and *Ae. aegypti*, by interfering with the locomotion and metabolic processes responsible for growth and development.

## 3. Conclusions and Future Perspectives

Besides its various properties and applications, chitosan, in its different formulations, is also known as a supportive tool (generally as a matrix for controlled-release compounds) for the control of many insect species harmful to agriculture, food, and veterinary and public health.

Among the different formulations of chitosan (such as plain chitosan, chitosan coating and film, chitosan and nematodes, chitosan chemical modification, chitosan nanoparticles loaded with EOs, insecticides, and different metabolites), chitosan added to EOs used as a coating film proved to be a promising technology to protect fruit, vegetables, and foodstuffs against insect pests and, in most cases, preserved the organoleptic characteristics and increased the shelf-life of fresh food products. This technique can be used to develop protective food coatings to discourage attack and oviposition by insect pests but also to create an active packaging material that could prevent fungi infestation and extend the shelf-life of products.

As reported in many articles, the nanoencapsulation of EOs in a chitosan matrix is also a promising formulation against insect pests. In fact, nanoparticles exhibit greater mobility than plain materials such as EOs or insecticides, resulting in better penetration into insect tissues and increasing insecticidal activity [103]. Moreover, chitosan-based nanoparticles showed an increase in the release time of EOs or insecticides compared to the starting products, allowing protection over time even at lower concentrations. In particular, chitosan nanoparticle formulations loaded with EOs could be considered the most promising EOs applications thanks to their eco-friendliness, effectiveness, protection over time, and delayed-release effect.

Unfortunately, the application of chitosan is limited by its poor solubility, which can be, however, improved using chemical modification by, for instance, introducing hydrophilic groups into the macromolecular chain of chitosan. Indeed, the chitosan derivatives NAC, NBC, and OAC with alkyl, benzyl, and acyl groups are more soluble than simple chitosan and exhibit, above all, higher insecticidal activity than plain chitosan. Among the different types of modified chitosan investigated, such as oligosaccharides and chitosan–metal complexes, some could be considered suitable in terms of applicability, effectiveness, and eco-friendliness.

In conclusion, despite the positive results reported in this paper and available in the recent literature, there are still a limited number of studies on the effect of various chitosan formulations on insect pests. With this review, we hope to provide readers with up-to-date information on the use and potential applications of chitosan-based formulations as pest control tools and stimulate interest in this promising substance.

## Figures and Tables

**Figure 1 insects-14-00949-f001:**
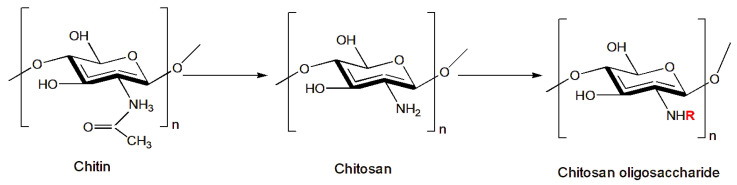
Formation of chitosan oligosaccharide (COS). R is replaced with H or an acetyl group depending on the degree of deacetylation (DD).

**Figure 2 insects-14-00949-f002:**
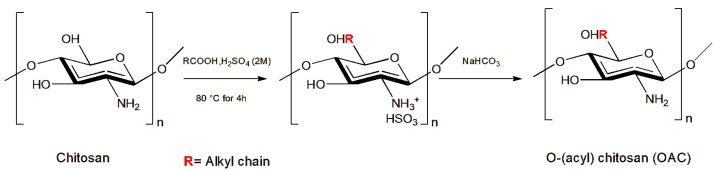
Synthesis of OAC. Redesigned from [38].

**Figure 3 insects-14-00949-f003:**
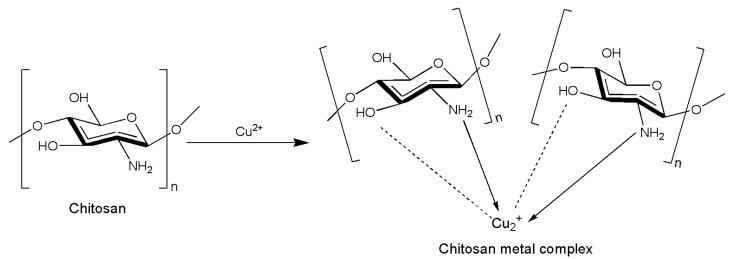
Possible formation of a chitosan–metal complex. Redesigned from Wang et al. [92].

**Figure 4 insects-14-00949-f004:**
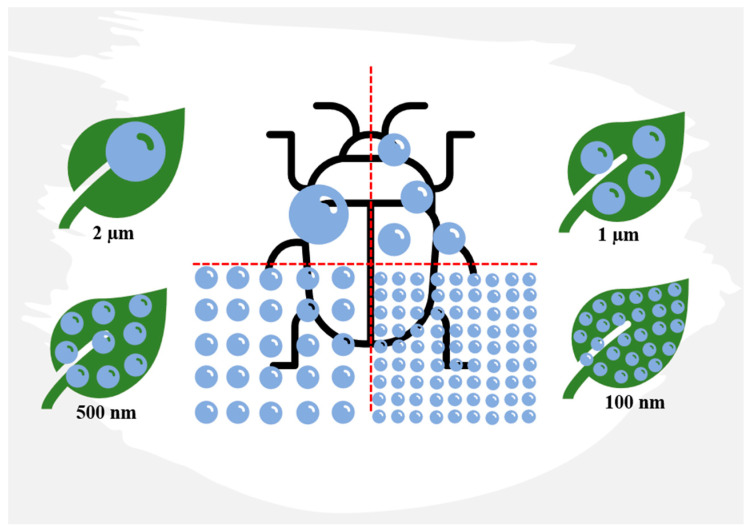
The ratio of particle size to the surface area to be treated. The smaller the particle size, the better the coverage, permeability, and bioavailability of the active ingredient as a tool for pest control. Redesigned from An et al. [94].

**Table 1 insects-14-00949-t001:** Chitosan formulations as insect pest control strategies in the literature from the last 20 years (from 2003 to 2023).

Chitosan Formulation	Insect Order	Insect Species/Family	Target Instars	References
Plain chitosan incorporated into a diet	Diptera	*Musca domestica* (Muscidae)	Adults	[22]
		*Tabanus nigrovittatus* (Tabanidae)	Adults	[22]
		*Phormia regina* (Calliphoridae)	Adults	[22]
Plain chitosan added to paper and wood	Blattodea	*Reticulitermes virginicus* (Rhinotermitidae)	Adults	[23]
		*Reticulitermes flavipes* (Rhinotermitidae)	Adults	[24]
Plain chitosan coating	Diptera	*Anastrepha ludens* (Tephritidae)	Larvae	[25]
		*Anastrepha obliqua* (Tephritidae)	Adults	[26]
Chitosan coating with EOs	Lepidoptera	*Plutella xylostella* (Plutellidae)	Adults	[27]
	Hemiptera	*Myzus persicae* (Aphididae)	Adults	[27]
	Diptera	*Bactrocera carambolae* (Tephritidae)	Adults	[28]
		*Calliphora vomitoria* (Calliphoridae)	Adults	[29]
	Coleoptera	*Acanthoscelides obtectus* (Bruchidae)	Adults	[30]
		*Sitophilus oryzae* (Curculionidae)	Adults	[31]
Chitosan coating in active packaging	Coleoptera	*Sitophilus zeamais* (Curculionidae)	Adults	[32]
Chitosan with nematodes	Coleoptera	*Ryncgophorus ferrugineus* (Curculionidae)	Larvae	[33,34]
Chitosan oligosaccharides	Lepidoptera	*Helicoverpa armigera* (Noctuidae)	Larvae	[35]
		*Plutella xylostella* (Plutellidae)	Larvae	[35]
	Hemiptera	*Sitobion avenae* (Aphididae)	Adults	[33]
		*Metopolophium dirhodum* (Aphididae)	Adults	[35]
		*Myzus persicae* (Aphididae)	Adults	[35]
		*Hyalopterus pruni* (Aphididae)	Adults	[35]
		*Aphis gossypii* (Aphididae)	Adults	[35]
		*Rhopalosiphum padi* (Aphididae)	Adults	[35]
Chitosan derivatives	Lepidoptera	*Spodoptera littoralis* (Noctuidae)	Larvae	[36,37,38]
Chitosan–metal complexes	Lepidoptera	*Spodoptera littoralis* (Noctuidae)	Larvae	[39]
	Hemiptera	*Aphis nerii* (Aphididae)	Adults	[39]
Chitosan nanoparticles loaded with EOs	Coleoptera	*Tribolium castaneum* (Tenebrionidae)	Adults	[20,21,40,41,42]
		*Sitophilus oryzae* (Curculionidae)	Adults	[21]
		*Oryzaephilus surinamensis* (Silvanidae)	Adults	[43]
		*Carpophilus hemipterus* (Nitidulidae)	Adults	[44]
	Diptera	*Anopheles stephensi* (Culicidae)	Larvae	[45]
		*Culex pipiens* (Culicidae)	Larvae	[46]
		*Aedes aegypti* (Culicidae)	Larvae	[19,47]
		*Musca domestica* (Muscidae)	Adults	[48]
	Hemiptera	*Bemisia tabaci* (Aleyrodidae)	Adults	[49]
Chitosan nanoparticles loaded with agrochemicals	Lepidoptera	*Helicoverpa armigera* (Noctuidae)	Larvae	[18]
		*Spodoptera litura* (Noctuidae)	Larvae	[50]
	Diptera	*Drosophila melanogaster* (Drosophilidae)	Adults/ larvae	[51]
Chitosan-g-poly acrylic acid nanoparticles	Hemiptera	*Aphis gossypii* (Aphididae)	Adults	[52]
	Coleoptera	*Cassida vittata* (Chrysomelidae)	Larvae	[53]
Myristic acid chitosan nanoparticles	Coleoptera	*Sitophilus granarius* (Curculionidae)	Adults	[54,55]
		*Tribolium confusum* (Tenebrionidae)	Adults	[54,55]
Chitosan nanoparticles RNAi	Diptera	*Anopheles gambie (* *Culicidae)*	Larvae	[56,57]
		*Aedes aegypti* (Culicidae)	Larvae	[57,58]

## Data Availability

Data are available on request from the corresponding authors.

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
