# Peer review of "Chitosan as a Control Tool for Insect Pest Management: A Review"

_insects, 2023, doi:10.3390/insects14120949_

Round 1

Reviewer 1 Report

Comments and Suggestions for Authors

The ppt version 1 of the manuscript is attached. Comments corrections and suggestions appear in highlighted sections of the manuscript.

Figure 1. Please follow comment on the nature of “R”.

Figures 2 and 3 (Updated version) please put brackets in all structures to indicate that they are part of a polymer.

Figure 3 apparently resulted from copy –paste and rotating the upper structure. This process results in the inversion of functional groups (such as –NH2 in this case). Please correct.

Figures 1, 2, & 3. To ensure uniformity please use a drawing program such as ChemDraw and select:

File à Apply document settings from à then select a setting (eg. ACS Document 1996)

Then select the figure and select “Object” à Scale à Press “Scale by” à 75%

That will give you the size and structure type used in most journals.

Comments on the Quality of English Language

The major drawback of the manuscript is that it is written in English, however, there are many unclear sections due to the wrong syntax used. The entire manuscript has to be reviewed by a native English speaker; that would result in a vast improvement of the paper.

Author Response

Thank you very much for the revision of this manuscript. You can find the detailed responses in the attached file and the modifications highlighted in track changes in the re-submitted file.

Reviewer 2 Report

Comments and Suggestions for Authors

The authors of this review explore the potential applications of all types of chitosan formulations, from the simplest to the most complex, as a tool for pest control. It is an interesting summary that can be published after the following edits:

Chitosan is currently permitted in EU countries as a so-called Basic substance. This fact should be mentioned and discussed (see e.g. DOI: 10.3390/horticulturae7100366; DOI: 10.3390/horticulturae9111220), because it mainly has fungicidal and eliciting effects. This fact should be part of the discussion.

Table 1 (or add a new table) should be supplemented with information regarding: application method (simplified information on dose or concentration, insect instars and application method) and observed efficacy (most important findings). This will make the table more complex and interesting. 

Author Response

(The authors gave the same response as above.)

Reviewer 3 Report

Comments and Suggestions for Authors

This manuscript can be published in the journal. The information on chitosan will be available for researchers in biochemical or entomophysiological study areas. Suggestions to improve the manuscript are given here.

1. English

English should be edited by a native English speaker before the submission of a revised manuscript. In particular, the usage of words and phrasing need to be improved.

2. Title

The title of this manuscript sounds like an original research paper. The title should clearly represent what is reported in the paper. The authors should change the title as such.

Table 1

As the authors mention (L81), this table shows the chitosan formulations which are grouped as simple to complicated. It would be helpful for readers to understand the table, if the authors provided the definition for the complexity and distinguish simple and complicated formulations.

L87

The word “harmful” is rather ambiguous. Give examples of insects assumed here.

L97

Describe how the microbiota would be changed and what results would be led in homeostasis by the changes.

L137

What does EO indicate? Any words should be spelled out when they are referred to first.

L273-281 and L289 – 297

Similar matters are mentions in these two paragraphs. One of them should be included in the other.

L363-365

The authors suggest in this sentence the “double” role of chitosan, insecticide and attractant. However, this suggestion may be risky. Chitosan in higher concentrations can achieves lethal effects on insects, while working as attractants at lower concentrations. This indicates that chitosan applied in sufficiently high concentration for the purpose of insect control may attract more insects later when the concentration is lowered to such a level as it works for attractant. The authors should refer to this risk.

L417

The genus of the latter species, Aedes, should be spelled out, since it is different from the former one, Anopheles. If the genus of the latter is abbreviated, readers may simply think these two species are congeneric.

Comments on the Quality of English Language

It is recommended for the authors to ask a native English speaker to edit the revised manuscript before the resubmission.

Author Response

(The authors gave the same response as above.)

Round 2

Reviewer 2 Report

Comments and Suggestions for Authors

The authors revised the manuscript according to the reviewers' comments. I have no further comments.